



# Lossy Checkpoint Compression in Full Waveform Inversion

Navjot Kukreja[1], Jan Hückelheim[2], Mathias Louboutin[3], John Washbourne[4], Paul H.J. Kelly[5], and
Gerard J. Gorman[1]

[1]Department of Earth Science and Engineering, Imperial College London
[2]Argonne National Laboratory
[3]Georgia Institute of Technology
[4]Chevron Corporation
[5]Department of Computing, Imperial College London

**Correspondence:** Navjot Kukreja (nkukreja@imperial.ac.uk)

**Abstract.** This paper proposes a new method that combines checkpointing methods with error-controlled lossy compression for large-scale high-performance Full-Waveform Inversion (FWI), an inverse problem commonly used in geophysical exploration. This combination can significantly reduce data movement, allowing a reduction in run time as well as peak memory.

In the Exascale computing era, frequent data transfer (e.g., memory bandwidth, PCIe bandwidth for GPUs, or network) is
the performance bottleneck rather than the peak FLOPS of the processing unit.

Like many other adjoint-based optimization problems, FWI is costly in terms of the number of floating-point operations, large memory footprint during backpropagation, and data transfer overheads. Past work for adjoint methods has developed checkpointing methods that reduce the peak memory requirements during backpropagation at the cost of additional floating-point computations.

Combining this traditional checkpointing with error-controlled lossy compression, we explore the three-way tradeoff between memory, precision, and time to solution. We investigate how approximation errors introduced by lossy compression of the forward solution impact the objective function gradient and final inverted solution. Empirical results from these numerical experiments indicate that high lossy-compression rates (compression factors ranging up to 100) have a relatively minor impact on convergence rates and the quality of the final solution.

**Keywords.** Lossy compression, Full waveform inversion, checkpointing, memory

## 1 Introduction

Full-waveform inversion (FWI) is an adjoint-based optimization problem used in seismic imaging to infer the earth's subsurface structure and physical parameters (Virieux and Operto, 2009). The compute and memory requirements for this and similar PDE-constrained optimization problems can readily push the world's top supercomputers to their limits. Table 1 estimates the
computational requirements of an FWI problem on the SEAM Model (Fehler and Keliher, 2011). Although the grid-spacing and timestep interval depends on various problem-specific factors, we can do a back-of-the-envelope calculation to appreciate the scale of FWI. To estimate the number of operations per grid point, we use a variant of Equation 1 called TTI (Zhang



et al., 2011), which is commonly used today in commercial FWI. Such a problem would require almost 90 days of continuous execution at 1 PFLOP/s. The memory requirements for this problem are also prohibitively high. As can be seen in Table 1, the

gradient computation step is responsible for this problem's high memory requirement, and the focus of this paper is to reduce that requirement.

The FWI algorithm is explained in more detail in Section 2. It is important to note that despite the similar terminology, the checkpointing we refer to in this paper is not done for resilience or failure recovery. This is the checkpointing from automatic-differentiation theory, with the objective of reducing the memory footprint of a large computation by trading recomputation for

storage.

| Description | Number | Peak Memory | No. of Flops |
|---|---|---|---|
| Single grid point (TTI) | 1 | 8 bytes | 6300 |
| Complete grid | $1000 \times 1000 \times 1000$ | 8GB | $6.3 \times 10^{12}$ |
| Forward propagation | 10000 | 24GB | $6.3 \times 10^{16}$ |
| Gradient Computation | 2 (FW+REV)[1] | 80TB | $1.26 \times 10^{17}$ |
| Shots | 10000 | 80TB | $1.26 \times 10^{21}$ |
| Optim. Iterations | 20 | 80TB | $2.52 \times 10^{22}$ |

**Table 1.** Estimated computational requirements of a Full-Waveform Inversion problem based on the SEAM model (Fehler and Keliher, 2011), a large scale industry standard geophysical model that is used to benchmark FWI. Note that real-world FWI problems are likely to be larger. [1]A gradient computation involves a forward simulation followed by a reverse/adjoint computation. For simplicity we assume the same size of computation during the forward/adjoint pass.

## 1.1    FWI and other similar problems

FWI is similar to other inverse problems like brain-imaging (Guasch et al., 2020), shape optimization (Jameson et al., 1998), and even training a neural network. When training a neural network, the activations calculated when propagating forward along the network need to be stored in memory and used later during backpropagation. The size of the corresponding computation

in a neural network depends on the depth of the network and, more importantly, the input size. We assume the input is an image for the purpose of this exposition. For typical input image sizes of less than $500 \times 500$ px, the computation per data point is relatively (to FWI) small, both in the number of operations and memory required. This is compensated by processing in *mini-batches*, where multiple data points are processed at the same time. This batch dimension's size is usually adjusted to *fill up* the target hardware to its capacity (and no more). This is the standard method of managing the memory requirements of a

neural network training pipeline. However, for an input image that is large enough, or a network that is deep enough, it is seen that the input image, network weights, and network activations together require more memory than available on a single node, even for a single input image (*batchsize* = 1 ). We previously addressed this issue in the context of neural networks (Kukreja et al., 2019b). In this paper we address the same issue for FWI.



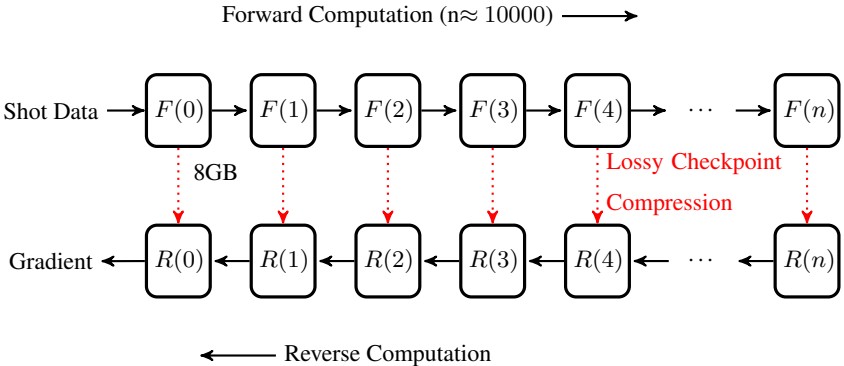

**Figure 1.** An illustration of the approach presented in this paper. Checkpoints are compressed using lossy compression to combine lossy compression and the checkpoint-recompute strategies.

Many algorithmic optimizations/approximations are commonly applied to reduce the computational load from the numbers calculated in Table 1. These optimizations could either reduce the number of operations or the amount of memory required. In this paper, we shall focus on the high memory footprint of this problem. One standard approach is to save the field at only the boundaries and reconstruct the rest of the field from the boundaries to reduce the memory footprint. However, the applicability of this method is limited to time-reversible PDEs. Although Equation 1 itself is time-reversible, most of the variations used in practice are not. For this reason, we do not discuss this method in this paper.

A commonly used method to deal with the problem of this large memory footprint is domain-decomposition over MPI, where the computational domain is split into subdomains over multiple compute nodes to use their memory. However, this strategy is limited by the parallel efficiency of the forward model and backpropagation, which can drop quickly as the subdomain's size per MPI rank decreases and the corresponding cost of interprocess communication increases.

Another notable approach that significantly reduces the amount of memory is frequency-domain methods. However, this method does not scale as well as time-domain methods (Knibbe et al., 2014).

Hybrid methods that combine time-domain methods, as well as frequency-domain methods, have also been tried (Witte et al., 2019b). However, this approach can be challenging because the application code must decide the user's discrete set of frequencies to achieve a target accuracy.

In the following subsections, we discuss three techniques that are commonly used to alleviate this memory pressure - namely numerical approximations, checkpointing, and data compression. The common element in these techniques is that all three solve the problem of high memory requirement by increasing the operational intensity of the computation - doing more computations per byte transferred from memory. With the gap between memory and computational speeds growing wider as we move into the exaflop era, we expect to use such techniques to increase moving forward.





## 1.2 Approximate methods

There has been some recent work on alternate floating-point representations (Chatelain et al., 2019), although we are not aware of this technique being applied to FWI. Within FWI, many approximate methods exist, including *On-the-fly Fourier transforms* (Witte et al., 2019a). However, it is not clear whether this method can provide fine-tuned bounds on the solution's accuracy. In contrast, other completely frequency-domain formulations can provide clearer bounds (van Leeuwen and Herrmann, 2014), but as previously discussed, this comes at the cost of a much higher computational complexity. In this paper, we restrict ourselves

to time-domain approaches only.

Another approximation commonly applied to reduce the memory pressure in FWI in the time domain is subsampling. Here, the computation for Equation 4 is split into two parts where $\nabla\Phi_s(\mathbf{m})$ is only calculated for one in $n$ evaluations of $v$. This reduces the memory footprint by a factor of $n$, since only one-in-$n$ values of $u$ need to be stored. The Nyquist theorem is commonly cited as the justification for this sort of subsampling. However, the Nyquist theorem only provides a lower bound

on the error - it is unclear whether an upper bound on the error has been established on this method. Although more thorough empirical measurements of the errors induced in subsampling have been done before (Louboutin* and Herrmann, 2015), we do a brief empirical study in Section 4.7 as a baseline to compare the error with our method.

## 1.3 Checkpointing

Instead of storing the value of $u$ from Equation 1 at every timestep during the initial forward computation, it is possible to store

$u$ at a subset of the timesteps only. During the following computation corresponding to Equation 4, if a value of $u$ is required at a timestep that was not stored, it can be recovered by restarting the forward computation from the last available timestep. This is commonly known as checkpointing. Algorithms have been developed to define the optimal checkpointing schedule involving forward, store, backward, load, and recompute events under different assumptions (Griewank and Walther, 2000; Wang et al., 2009; Aupy and Herrmann, 2017). This technique has also been applied to FWI-like computations (Symes, 2007).

In previous work, we introduced the open-source software pyRevolve, a Python module that can automatically manage the checkpointing strategies under different scenarios with minimal modification to the computations code (Kukreja et al., 2018). In this paper, we extend pyRevolve to integrate lossy compression.

The most significant advantage of checkpointing is that the numerical result remains unchanged by applying this technique. Note that we will shortly combine this technique with lossy compression which might introduce an error, but checkpointing

alone is expected to maintain bitwise equivalence. Another advantage is that the increase in run time incurred by the recomputation is predictable.

## 1.4 Data Compression

Compression or bit-rate reduction is a concept originally from signal processing. It involves representing information in fewer bits than the original representation. Since there is usually some computation required to go from one representation to another,

compression can be seen as a memory-compute tradeoff.



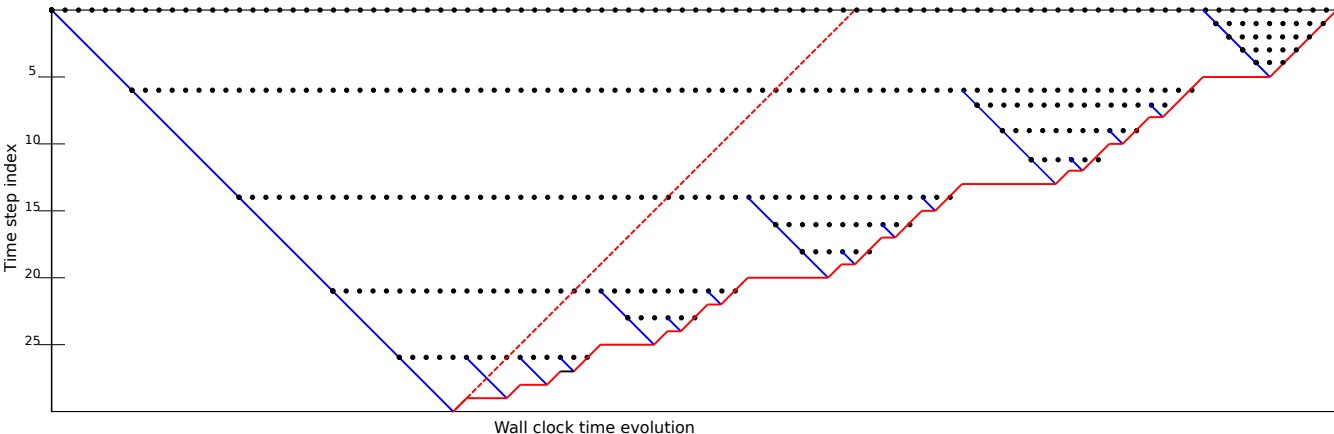

**Figure 2.** Schematic of the checkpointing strategy. Wall-clock time is on the horizontal axis, while the vertical axis represents simulation time. The blue line represents forward computation. The dotted red line represents how the reverse computation would have proceeded after the forward computation, had there been enough memory to store all the necessary checkpoints. Checkpoints are shown as the black dots. The reverse computation under the checkpointing strategy is shown as the solid red line. It can be seen that the reverse computation proceeds only where the results of the forward computation are available. When not available, the forward computation is restarted from the last available checkpoint to recompute the results of the forward.

Perhaps the most commonly known and used compression algorithm is ZLib (from GZip) (Deutsch and Gailly, 1996). TZLib is a lossless compression algorithm, i.e., the data recovered after compressing-decompressing is an exact replica of the original data before compression. Although ZLib is targeted at text data, which is one-dimensional and often has predictable repetition, other lossless compression algorithms are designed for other kinds of data. One example is FPZIP (Lindstrom et al., 2017), 100  which is a lossless compression algorithm for multidimensional floating-point data.

For floating-point data, another possibility is lossy compression, where the compressed-decompressed data is not exactly the same as the original data, but a close approximation. The precision of this approximation is often set by the user of the compression algorithm. Two popular algorithms in this class are SZ (Di and Cappello, 2016) and ZFP (Lindstrom, 2014).

Compression has often been used to reduce the memory footprint of adjoint computations in the past, including Boehm et al. 105  (2016); Marin et al. (2016). However, both these studies use hand-rolled compression algorithms specific to the corresponding task (wave propagation in the case of Boehm et al. (2016), fluid flow in the case of Marin et al. (2016)) and use the algorithm to compress the entire time history. In this paper we use a more general floating-point compression algorithm, and a combination of checkpointing and compression – both of which extend the viability of the method.

Cyr et al. (2015) performs numerical experiments to study the propagation of errors through an adjoint problem using 110  compression methods like PCA. However, they do not consider the combination of checkpointing and compression in a single strategy.

Floating-point can be seen as a *compressed representation* that is not entirely precise. However, the errors introduced by the floating-point representation are already accounted for in the standard numerical analysis as noise. The lossy compression of

fields is more subtle since the compression loss is pattern sensitive – beyond known numerical analysis. Hence we tackle it

empirically here.

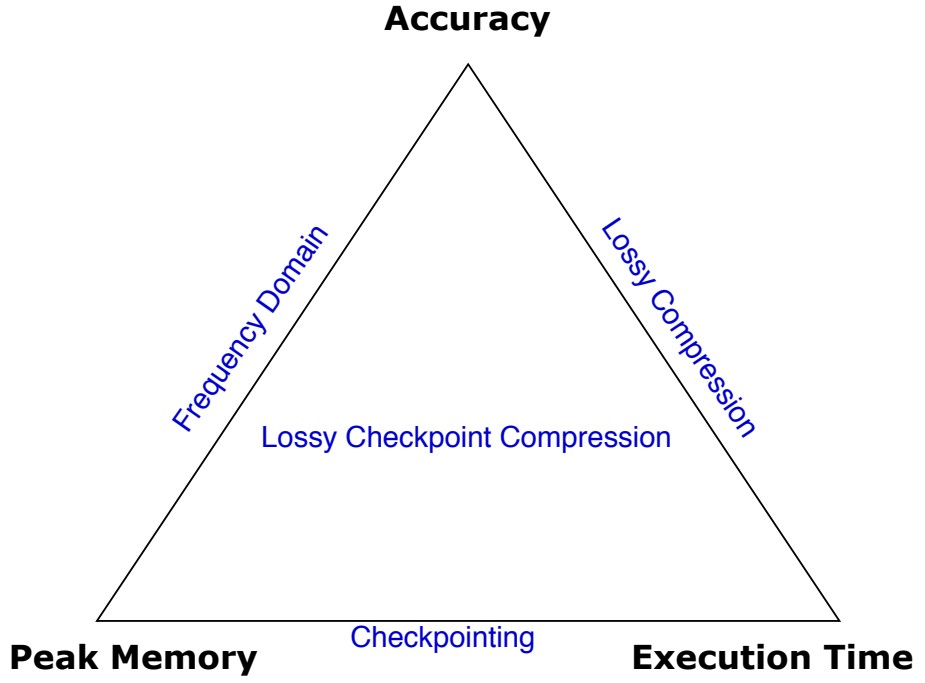

**Figure 3.** Schematic of the three-way tradeoff presented in this paper. With the use of checkpointing, it was possible to trade off memory and execution time (the horizontal line). With the use of compression alone, it was possible to trade off memory and accuracy. The combined approach presented in this work provides a novel three-way tradeoff.

### 1.5 Contributions

The last few sections discussed some existing methods that allow trade-offs that are useful in solving FWI on limited resources. While checkpointing allows a trade-off between computational time and memory, compression allows a trade-off between memory and accuracy. In this work, we wish to combine these approaches into one three-way trade-off.

In previous work (Kukreja et al., 2019a), we have shown that it is possible to accelerate adjoint-based computations, including FWI, by using lossy compression on the checkpoints. For a given checkpoint absolute error tolerance (*atol* ), compression may or may not accelerate the computation. The performance model from Kukreja et al. (2019a) helps us answer this question a priori, i.e., without running any computations.

In this paper, we evaluate how lossy compression impacts solver convergence and accuracy in FWI.

To this end, we conduct an empirical study of:





1. Propagation of errors when starting from a lossy checkpoint.

2. Effect of checkpoint errors on the gradient computation.

3. Effect of decimation/subsampling on the gradient computation.

4. Accumulation of errors through the stacking of multiple shots.

5. Effect of the lossy gradient on the convergence of FWI.

The rest of the paper is organized as follows. Section 2 gives an overview of FWI. This is followed by a description of our experimental setup in Section 3. Next, Section 4 discusses the results, followed by our conclusions.

## 2   Full Waveform Inversion

FWI is designed to numerically simulate a seismic survey experiment and invert for the earth parameters that best explain
the observations. In the physical experiment, a ship sends an acoustic impulse through the water by triggering an explosion. The waves created as a result of this impulse travel through the water into the earth's subsurface. The reflections and turning components of these waves are recorded by an array of receivers being dragged in tow by the ship. A recording of one signal sent and the corresponding signals received at each of the receiver locations is called a shot. A single experiment typically consists of  10000 shots.

Having recorded this collection of data ($d_{obs}$), the next step is the numerical simulation. This starts with a wave equation. Many equations exist that can describe the propagation of a sound wave through a medium - the choice is usually a trade-off between accuracy and computational complexity. We mention here the simplest such equation, that describes isotropic acoustic wave propagation:

$$m(x)\frac{\partial^2 u(t,x)}{\partial t^2} - \nabla^2 u(t,x) = q(t,x), \tag{1}$$

where $m(x) = \frac{1}{c^2(x)}$ is the squared slowness, $c(x)$ the spatially dependent speed of sound, $u(t,x)$ is the pressure wavefield, $\nabla^2 u(t,x)$ denotes the laplacian of the wavefield and $q(t,x)$ is a source term. Solving Equation 1 for a given $m$ and $q_s$ can give us the *simulated* signal that *would be* received at the receivers. Specifically, the *simulated data* can be written as:

$$\mathbf{d}_{\text{sim}} = \mathbf{P}_r \mathbf{u} = \mathbf{P}_r \mathbf{A}(\mathbf{m})^{-1} \mathbf{P}_s^\top \mathbf{q}_s \tag{2}$$

where $\mathbf{P}_r$ is the measurement operator that restricts the full wavefield to the receivers locations, $\mathbf{A}(\mathbf{m})$ is the linear operator
that is the discretization of Equation 1, and $\mathbf{P}_s$ is an operator that injects the source signal ($\mathbf{q}_s$) into the localized source positions.

Using this, it is possible to set up an optimization problem that aims to find the value of $m$ that minimizes the difference between the simulated signal ($d_{sim}$) and the observed signal ($d_{obs}$):

$$\min_m \phi_s(m) = \frac{1}{2}\|\mathbf{d}_{\text{sim}} - \mathbf{d}_{\text{obs}}\|_2^2. \tag{3}$$





This objective function $\phi_s(m)$ can be minimized using a gradient descent method. The gradient can be computed as follows:

$$\nabla\Phi_s(\mathbf{m}) = \sum_{t=1}^{n_t} \mathbf{u}[\mathbf{t}]\mathbf{v}_{tt}[\mathbf{t}] = \mathbf{J}^T\delta\mathbf{d} \qquad (4)$$

where $\mathbf{u}[\mathbf{t}]$ is the wavefield from Equation 1 and $\mathbf{v}_{tt}[\mathbf{t}]$ is the second-derivative of the adjoint field (Tarantola, 1984). The adjoint field is computed by solving an adjoint equation *backwards* in time. The appropriate adjoint equation is a result of the choice of

the forward equation. In this example, we chose the acoustic isotropic equation (Equation 1), which is self-adjoint. However, it is not always trivial to derive the adjoint equation corresponding to a chosen forward equation (Hückelheim et al., 2019). This adjoint computation can only be started once the forward computation (i.e. the one involving Equation 1) is complete. Commonly, this is done by storing the intermediate values of $u$ during the forward computation, then starting the adjoint computation to get values of $v$, and using that and the previously calculated $u$ to directly calculate $\nabla\Phi_s(\mathbf{m})$ in the same loop.

This need to store the intermediate values of $u$ during the forward computation is the source of the high memory footprint of this method.

## 3    Experimental setup

**Reference Problem**    We use Devito (Kukreja et al., 2016; Luporini et al., 2018) to build an acoustic wave propagation experiment. The velocity model was initialized using the SEG Overthrust model. This velocity model was then smoothed using

a Gaussian function to simulate a starting guess for a complete FWI problem. The original domain was surrounded by a 40 point deep absorbing boundary layer. This led to a total of $287 \times 881 \times 881$ grid points. This was run for $4000ms$ with a step of $1.75ms$, making 2286 timesteps. The spatial domain was discretized on a grid with a grid spacing of 20m, and the discretization was 16th-order in space and second-order in time. We used 80 shots for our experiments with the sources placed along the x-dimension, spaced equally and just under the water surface. The shots were generated by modeling

a Ricker source of peak frequency 8Hz. Following the method outlined in Peters et al. (2019), we avoid inverse crime by generating the shots using a variation of Equation 1 that includes density, while using Equation 1 for inversion. The gradient was scaled by dividing by the norm of the original gradient in the first iteration. This problem solved in double precision is what we shall refer to as the *reference problem* in the rest of this paper. Note that this reference solution itself has many sources of error, including floating-point arithmetic and the discretization itself.

**Evolution of compressibility**    We attempt to compress every timestep of the reference problem using the same compression setting and report on the achieved compression factor as a function of the timestep.

**Direct compression**    Based on the previous experiment, we choose a reference wavefield and compress it directly using a variety of compression settings. In this experiment, we report the errors comparing the lossy wavefield and the true reference wavefield.





**Forward propagation** In this experiment, we run the forward simulation for a few timesteps (about half the reference problem) and store it as a checkpoint. We then compress and decompress this through the lossy compression algorithm, getting two checkpoints - a reference checkpoint and a lossy checkpoint. We restart the simulation from each of these checkpoints and compare the two simulations' states and report on differences.

**Gradient Computation** In this experiment, we do the complete gradient computation, as shown in Figure 1 - once for the reference problem and a few different lossy settings. We report on the differences between these to show the propagation of errors.

**Stacking** In this experiment, we collate the gradient computed on multiple shots, i.e., all ten shots, and report the difference between the reference problem and the compressed version for this step.

**Convergence** In practice, FWI is run for only a few iterations at a time as a fine-tuning step interspersed with other imaging steps. Here we run a fixed number of FWI iterations (30) to make it easier to compare different experiments. To make this a practical test problem, we extract a 2D slice from the original 3D velocity model and run a 2D FWI instead of 3D. We compare the convergence trajectory with the reference problem and report.

**Subsampling** As a comparison baseline, we also use subsampling to reduce the memory footprint as a separate experiment and track the errors. The method is set up so that the forward and adjoint computations continue at the same time stepping as the reference problem above. However, the gradient computation is now not done at the same rate - it is reduced by a factor $f$. We plot results for varying $f$.

## 4 Results and Discussion

### 4.1 Evolution of compressibility

To understand the evolution of compressibility, we tried to compress each and every timestep of a simulation to observe the evolution of compressibility through the simulation. This is shown in Figure 4. It can be seen that in the beginning the field is highly compressible since it consists of mostly zeros. The compressibility is worst towards the end of the simulation when the wave has reached most of the domain.

Therefore we pick the last timestep as the reference for further experiments. A 2D cross section of this snapshot is shown in Figure 5.

### 4.2 Direct compression

To understand the direct effects of compression, we compressed the reference wavefield using a variety of absolute tolerance (*atol* ) settings and observed the errors incurred as a function of *atol* . The error is a tensor of the same shape as the original field and results from subtracting the reference field and the lossy field. Figure 6 shows the Peak Signal-to-Noise Ratio achieved for each *atol* setting. Figures A1 and A2 in the appendix show some additional norms for this error tensor.



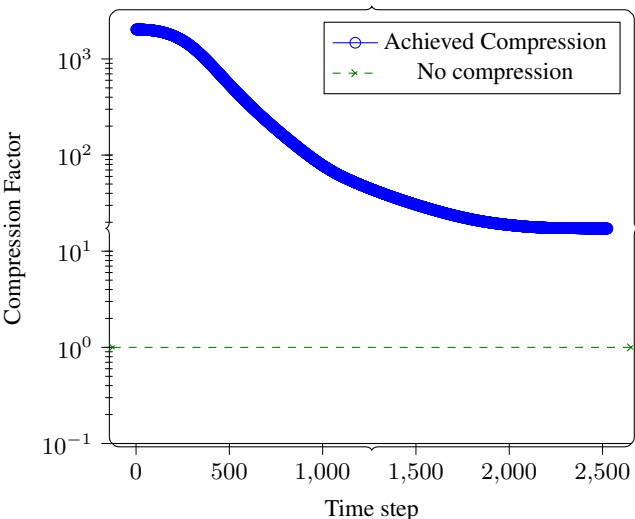

**Figure 4.** Evolution of compressibility through the simulation. We tried to compress every time step of the reference problem using an absolute error tolerance (*atol*) setting of $10^{-4}$. The Compression factor achieved is plotted here as a function of the timestep number. Higher is more compression. Dotted line represents *no compression*. We can see that the first few timesteps are compressible to 1000x - since they are mostly zeros. The achievable compression factor drops as the wave propagates through the domain and seems to stabilize to ~20x towards the end. We pick the last time step as the reference field for further experiments.

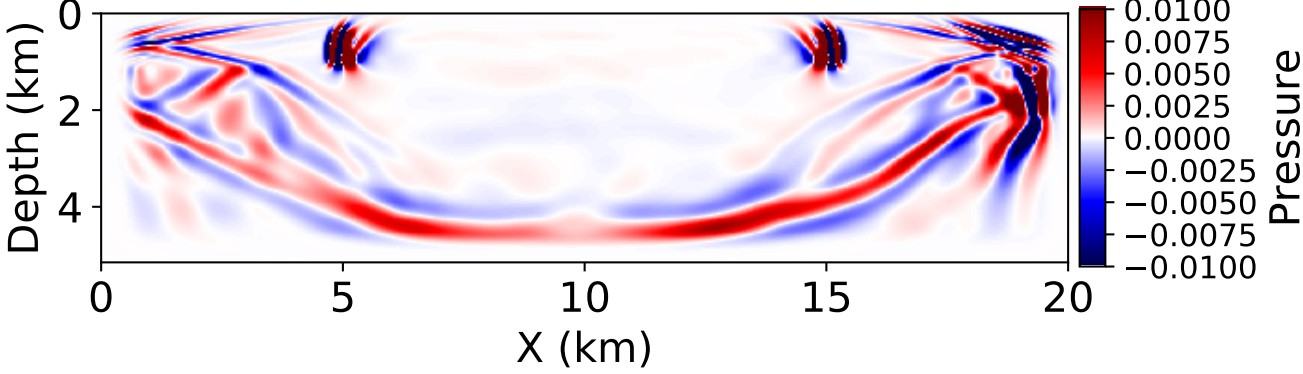

**Figure 5.** A 2D slice of the last time step of the reference solution. The wave has spread through most of the domain.





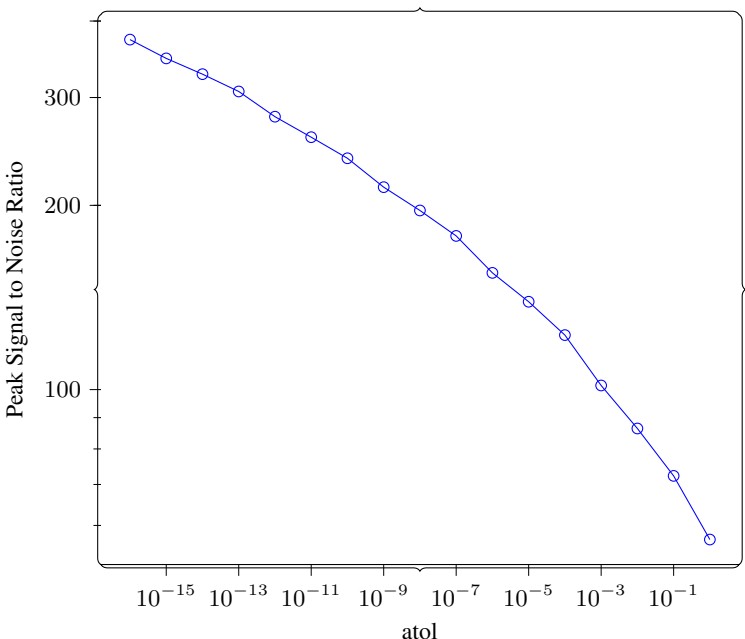

**Figure 6.** Direct compression: We compress the wave field at the last time step of the reference solution using different *atol* settings and report the Peak Signal to noise ratio (PSNR) achieved. Higher PSNR is lower error. The PSNR is very high for low absolute error tolerance (*atol*) and drops predictably as *atol* is increased. See figures A1 and A2 for more metrics on this comparison.

## 4.3 Forward propagation

Next, we ran the simulation for 500 steps and compressed the field's final state after these 500 steps. We then restarted the simulation from step 500, comparing the progression of the simulation restarted from the lossy checkpoint vs. a *reference* simulation that was started from the original checkpoint.

Figure 7 shows the evolution of various error metrics as a function of the number of timesteps evolved. The $L_\infty$ norm does not seem to be growing with the number of timesteps. This tells us that the maximum magnitude of pointwise error is not growing with the timesteps. However, We can see that the $L_2$ norm of the error is growing linearly. This could mean that the same magnitude of pointwise error is spreading to more parts of the domain - as part of the wave being modeled.

## 4.4 Gradient computation

Next, we measured the error in the gradient computation as a function of *atol*, assuming the same compression settings are used for all checkpoints.

Apart from showing that the error in the gradient remains almost constant with changing *atol*, Figure 10 also shows that the number of timesteps do not appear to change the error by much (for a constant number of checkpoints).



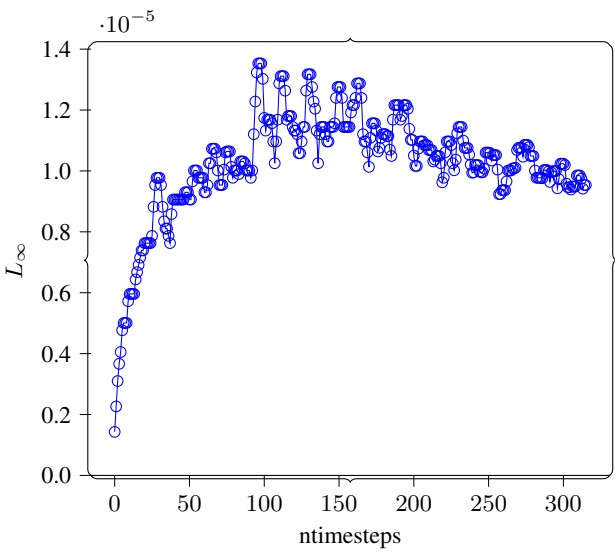

**Figure 7.** Forward propagation: We stop the simulation at half the number of timesteps as compared to the reference solution. We then compress the state of the wavefield at this point using $atol = 10^{-4}$. We then continue the simulation from the lossy checkpoint and compare with the reference version. Here we report the $L_\infty$ norm of the error between the wavefields of these two versions as a function of the number of timesteps evolved from this lossy checkpoint. Since $L_\infty$ does not grow by much, we can conclude that the maximum pointwise error is not growing in magnitude. See Figure 8 for $L_2$ norm.

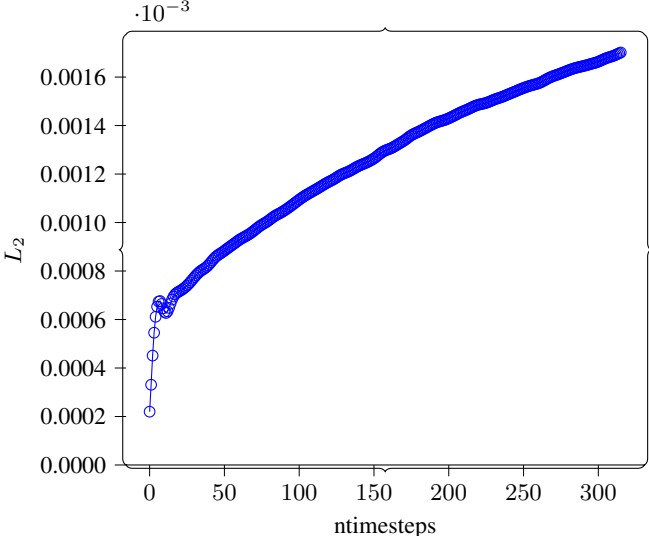

**Figure 8.** Forward propagation: Same experiment as described in Figure 7. Here we report the $L_2$ norm of the error between the wavefields of these two versions as a function of the number of timesteps evolved from this lossy checkpoint. The growing $L_2$ norm tells us that the average error is going up possibly because the error is spreading around the domain without increasing in magnitude.



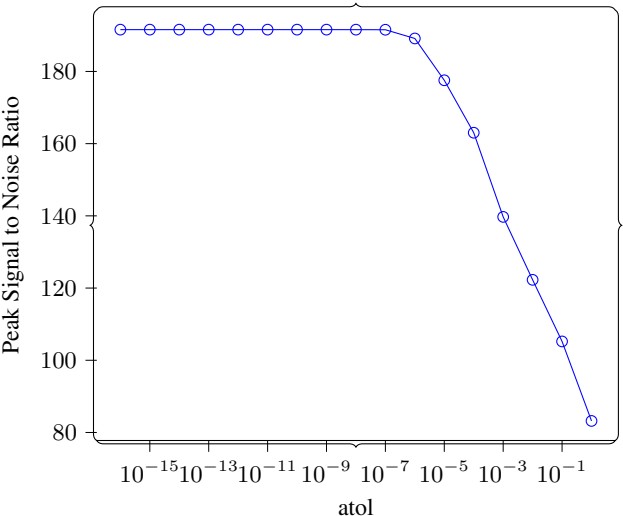

**Figure 9.** Gradient computation: In this experiment we carry out the full forward-reverse computation to get a gradient for a single shot, while compressing the checkpoints at different *atol* settings. This plot shows the PSNR of true vs lossy gradient as a function of *atol* on the lossy checkpoints. We can see that the PSNR remains unchanged until about $atol = 10^{-6}$ and is very high even at very high values of *atol* .

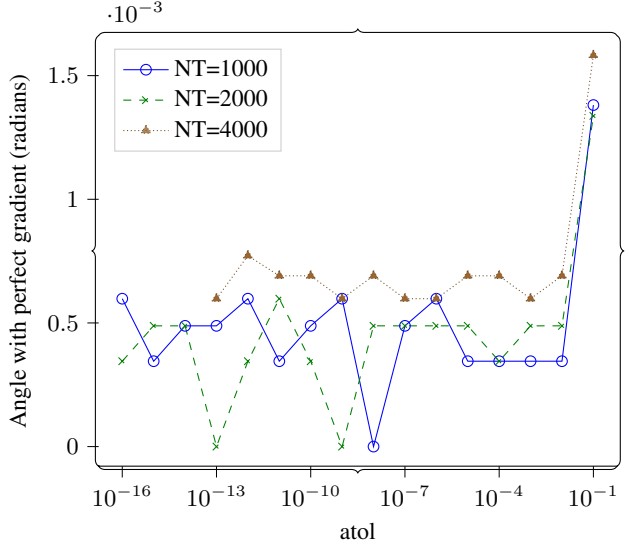

**Figure 10.** Gradient computation: Angle between the lossy gradient vector and the reference gradient vector (in radians) vs *atol* . If the lossy gradient vector was pointing in a significantly different direction as compared to the reference gradient, we could expect to see that on this plot. The angles are quite small. The number of timesteps do not affect the result by much. The results are also resilient to increasing *atol* up to $10^{-2}$ .





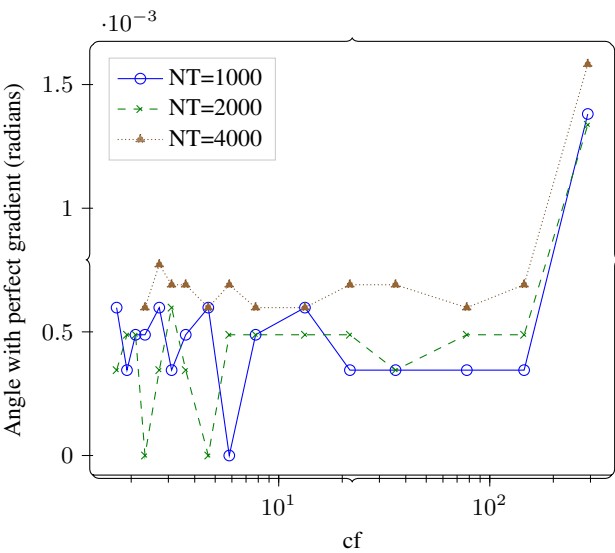

**Figure 11.** Gradient computation: Angle between the lossy gradient vector and the reference gradient vector (in radians) vs compression factor. Same experiment as Figure 10. Compression factors of over 100x do not seem to significantly distort the results either.

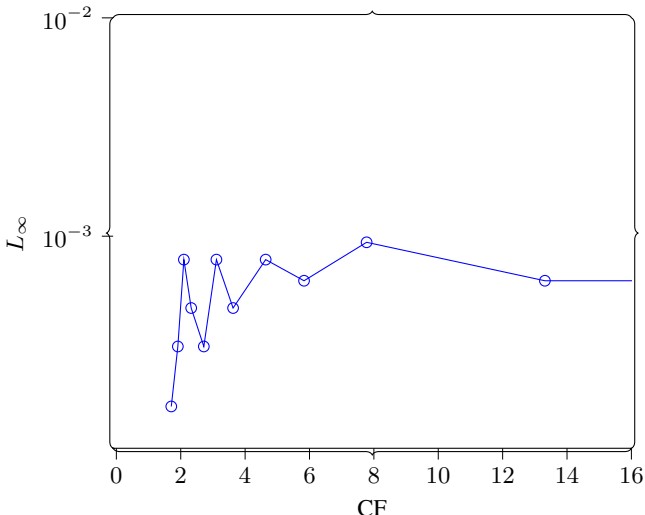

**Figure 12.** Gradient error: $L_\infty$ norm of the gradient error as a function of the achieved compression factor (CF). It can be seen that error is negligible in the range of CF up to 16. Compare this to subsampling in Figure 24. Note that we achieved much higher CF values as part of the experiment but cut the axis in this figure to make it comparable to Figure 24

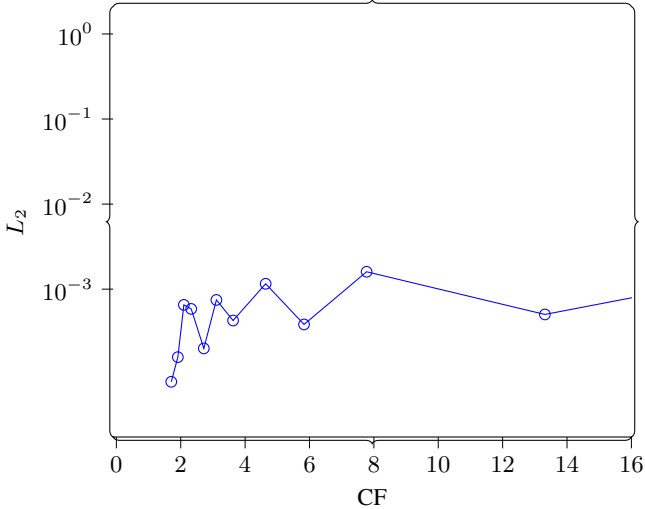

**Figure 13.** Gradient error: $L_2$ norm of the gradient error as a function of the achieved compression factor (CF). It can be seen that errors are negligible in the range of CF up to 16. Compare this to subsampling in Figure 24. Note that we achieved much higher CF values as part of the experiment but cut the axis in this figure to make it comparable to Figure 24

It can be seen from the plots that the errors induced in the checkpoint compression do not propagate significantly until the gradient computation step. In fact, the *atol* compression setting does not affect the error in the gradient computation until a cutoff point. It is likely that the cross-correlation step in the gradient computation is acting as an error-correcting step since the adjoint computation continues at the same precision as before - the only errors introduced are in the values from the forward computation used in the cross-correlation step (the dotted arrows in Figure 1).

### 4.5 Stacking

After gradient computation on a single shot, the next step in FWI is the accumulation of the gradients for individual shots by adding them into a single gradient. We call this stacking. In this experiment we studied the accumulation of errors through this stacking process. Figure 17 shows the error in the gradient computation (compared to a similarly processed reference problem) as a function of the number of shots.

This plot shows us that the errors across the different shots are not adding up and the cumulative error is not growing with the number of shots - except for the compression setting of $atol = 10^{-1}$, which is chosen as an example of unreasonably high compression.

### 4.6 Convergence

Finally, we measure the effect of an approximate gradient on the convergence of the FWI problem. For reference, Figure 19 shows the known true velocity model for this problem. Figure 19 shows the final velocity model after running a *reference* FWI





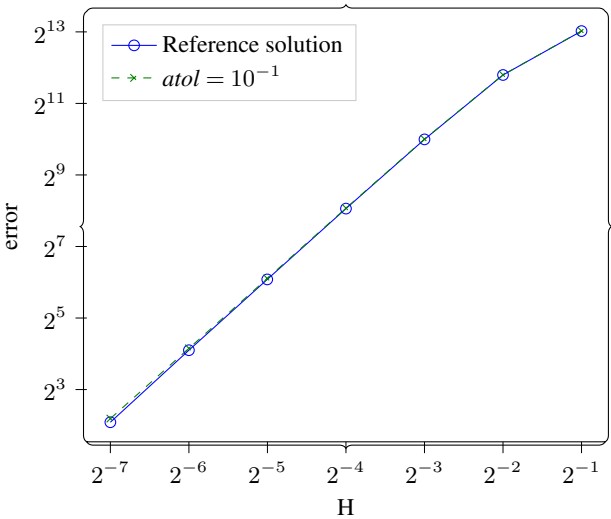

**Figure 14.** Gradient linearization: Comparison of gradient linearization errors for $atol = 10^{-1}$ vs reference solution. The horizontal axis represents a small linear perturbation to the velocity model and the vertical axis represents the error observed at that perturbation. The two curves in each of the plots follow each other so closely that they are indistinguishable. This confirms that the lossy gradient satisfies the Taylor linearization properties just as well as the reference gradient.

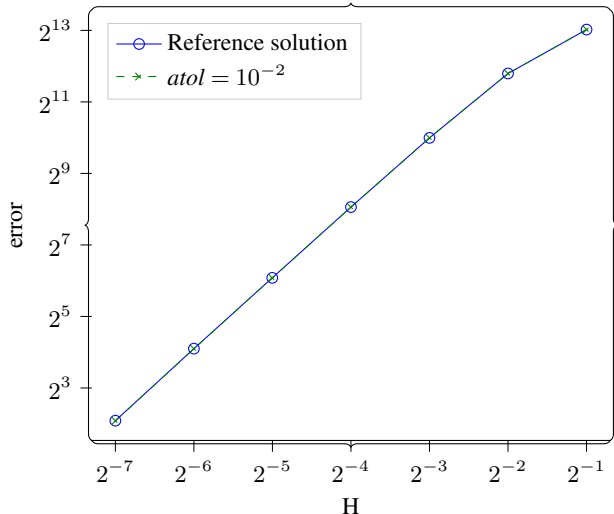

**Figure 15.** Gradient linearization: Comparison of gradient linearization errors for $atol = 10^{-2}$ vs reference solution. The horizontal axis represents a small linear perturbation to the velocity model and the vertical axis represents the error observed at that perturbation. The two curves in each of the plots follow each other so closely that they are indistinguishable. This confirms that the lossy gradient satisfies the Taylor linearization properties just as well as the reference gradient.



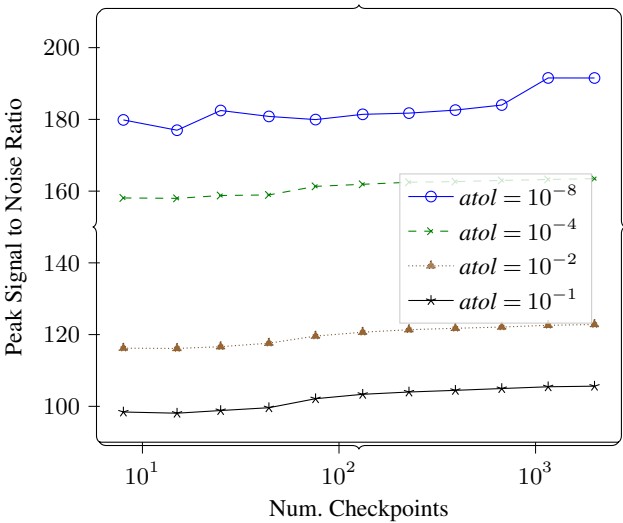

**Figure 16.** Gradient error: In this plot we measure the effect of varying number of checkpoints on the error in the gradient. We report PSNR of lossy vs reference gradient as a function of number of checkpoints, for four different compression settings.

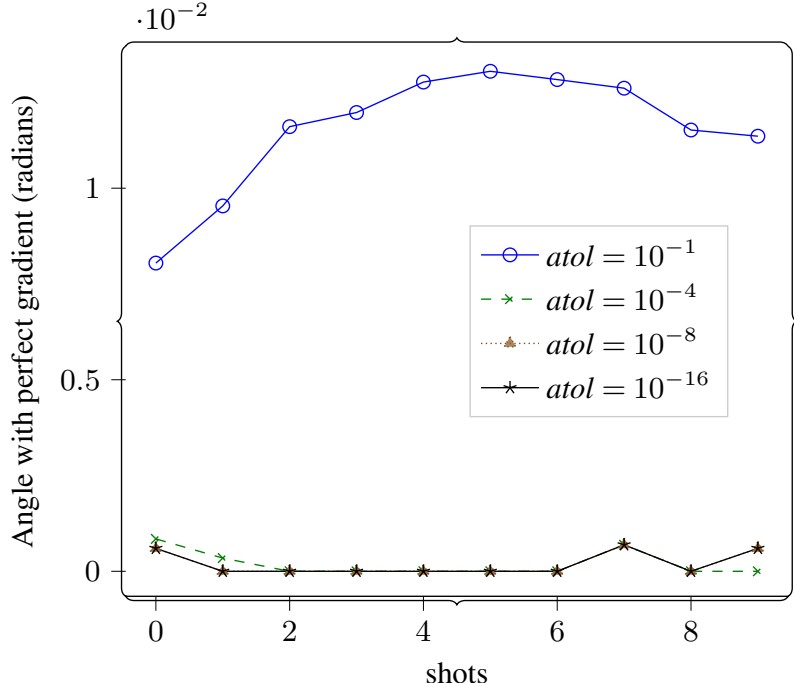

**Figure 17.** Shot stacking: The gradient is first computed for each individual shot and then added up for all the shots. In this experiment we measure the propagation of errors through this step. This plot shows that while errors do have the potential to accumulate through the step - as can be seen from the curve for $atol = 10^{-1}$, for compression settings that are useful otherwise, the errors do not accumulate significantly.



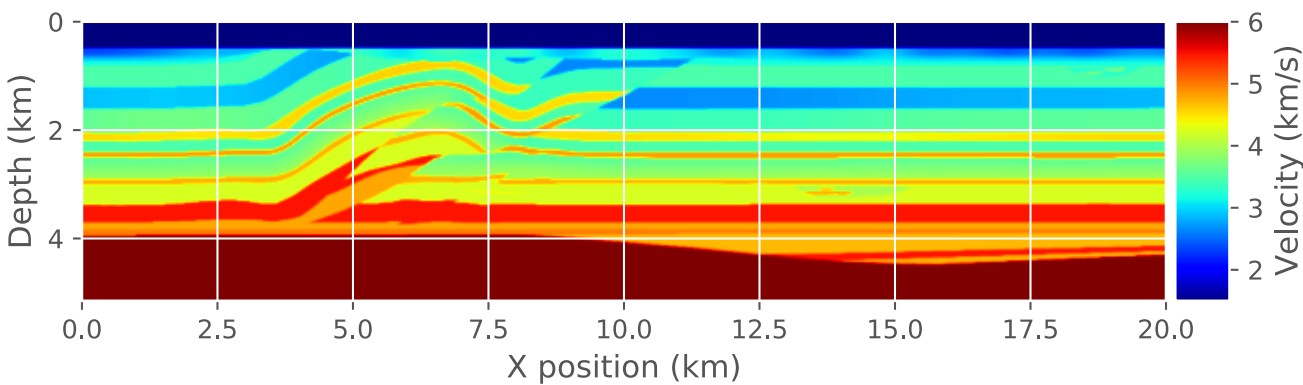

**Figure 18.** True solution for FWI

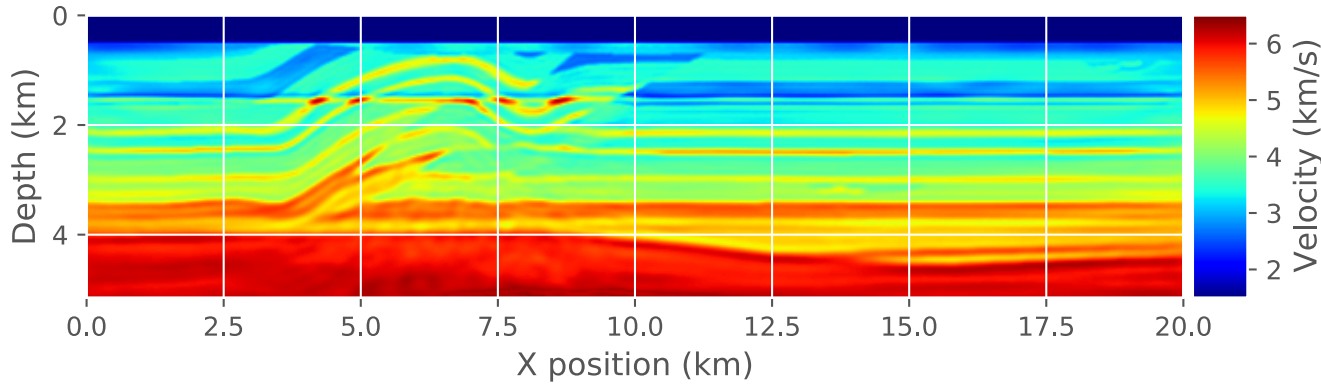

**Figure 19.** Reference solution for the complete FWI problem. This is the solution after running *reference* FWI for 30 iterations

for 30 iterations. Figure 22 shows the final velocity model after running FWI with compression enabled at different *atol* settings

- also for 30 iterations.

Figure 20 shows the convergence trajectory - the objective function value as a function of the iteration number. We show this convergence trajectory for 4 different compression settings. It can be seen that the compressed version does indeed follow a very similar trajectory as the original problem.

### 4.7 Subsampling

To compare our proposed method with subsampling - which is sometimes used in industry, we run an experiment where we use subsampling in time to reduce the memory footprint. Figure 24 shows some error metrics as a function of the compression factor $f$.





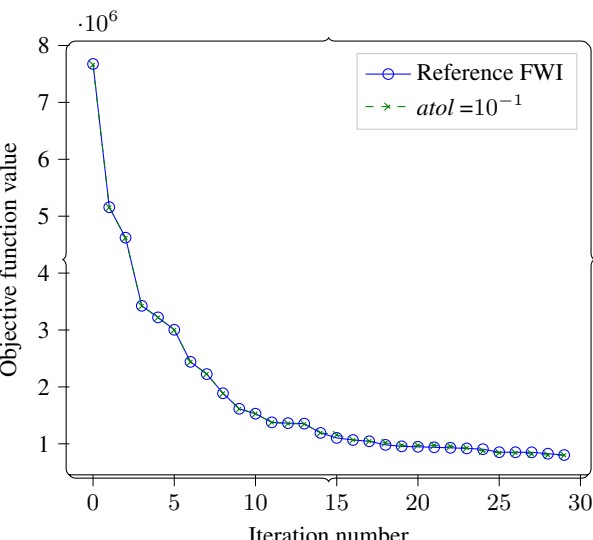

**Figure 20.** Convergence: As the last experiment, we run complete FWI to convergence (up to max 30 iterations). Here we show the convergence profiles for $atol = 10^{-1}$ vs the reference problem. The reference curve is so closely followed by the lossy curve that the reference curve is hidden behind.

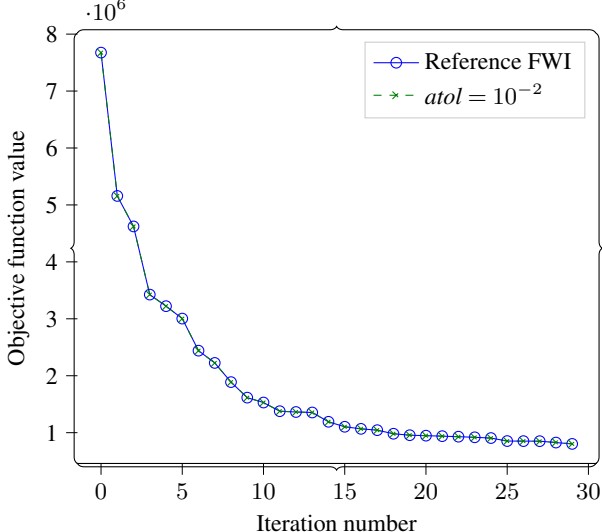

**Figure 21.** Convergence: As the last experiment, we run complete FWI to convergence (up to max 30 iterations). Here we show the convergence profiles for $atol = 10^{-2}$ vs the reference problem. The reference curve is so closely followed by the lossy curve that the reference curve is hidden behind.

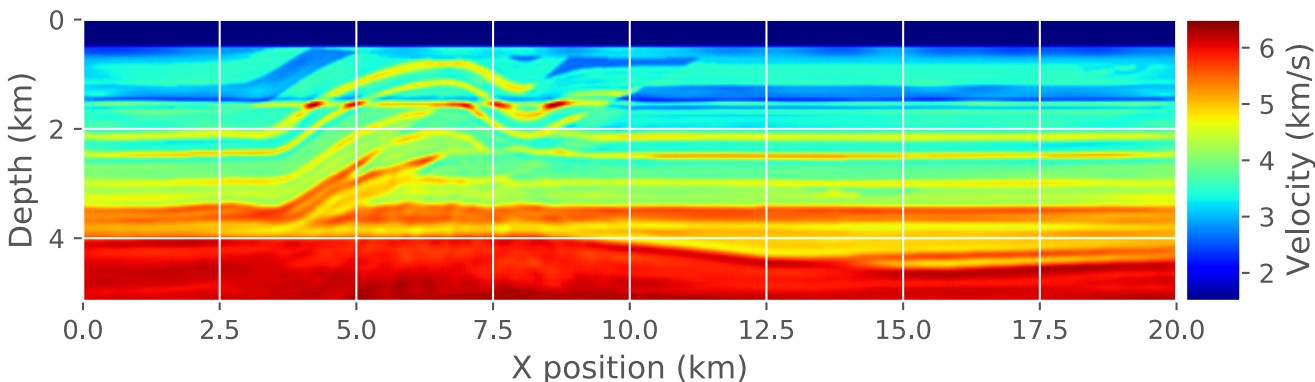

**Figure 22.** Final image after running FWI $atol = 10^{-1}$. It is visually indistinguishable from the reference solution in Figure 19.

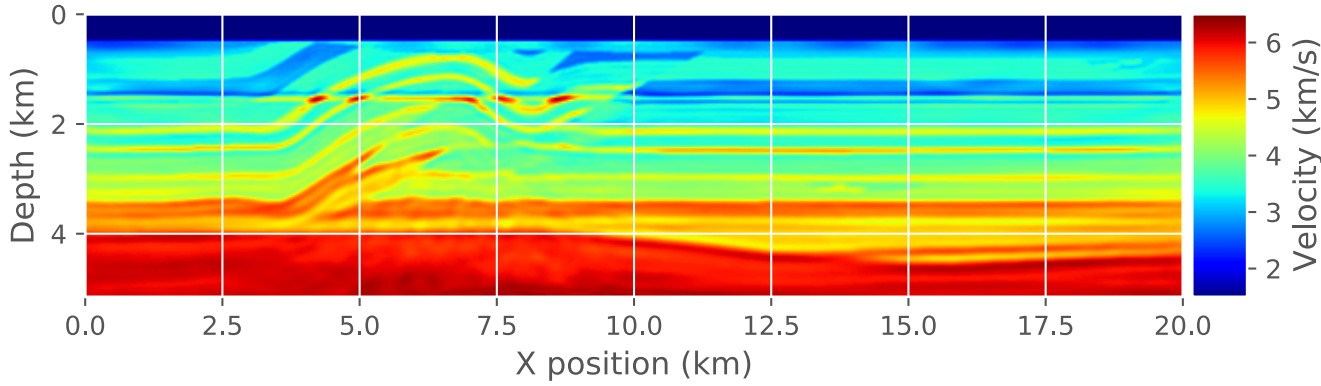

**Figure 23.** Final image after running FWI $atol = 10^{-2}$. It is visually indistinguishable from the reference solution in Figure 19.

Comparing Figure 24 with Figure 12, it can be seen that the proposed method produces significantly smaller errors for similar compression factors.

The results indicate that significant lossy compression can be applied to the checkpoints before the solution is adversary affected. This being an empirical study, we can only speculate on the reasons for this. We know that in the proposed method, the adjoint computation is not affected at all - the only effect is in the wavefield carried over from the forward computation to the gradient computation step. Since the gradient computation is a cross-correlation, we only expect correlated signals to grow in magnitude in the gradient calculation and when gradients are stacked. The optimization steps are likely to be error-correcting

as well since even with an approximate gradient ($atol > 4$), the convergence trajectory and the final results do not appear to change much - indicating that the errors in the gradient might be canceling out over successive iterations. There is even the possibility that these *errors* in the gradient introduce a kind of regularization. The number of checkpoints has some effect on the error - more checkpoints incur less error for the same compression setting - as would be expected.



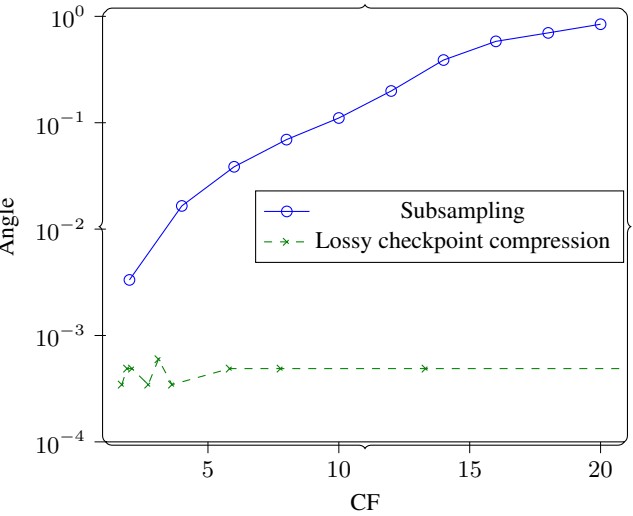

**Figure 24.** Subsampling: We set up an experiment with subsampling as a baseline for comparison. Subsampling is when the gradient computation is carried at a lower timestepping than the simulation itself. This requires less data to be carried over from the forward to the reverse computation at the cost of solution accuracy so is comparable to lossy checkpoint compression. This plot shows the angle between the lossy gradient and the reference gradient versus the compression factor $CF$ for this experiment. Compare this to the errors in Figure 12 that apply for lossy checkpoint compression.

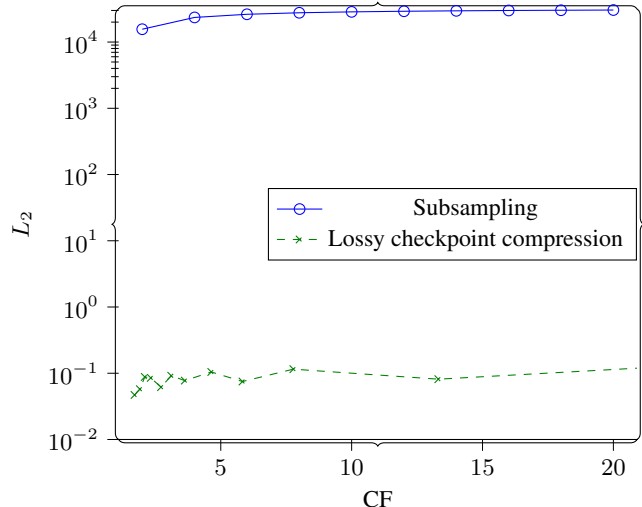

**Figure 25.** Subsampling: Same experiment as in Figure 24. This plot shows $L_2$ norm of gradient error versus the compression factor $CF$ for this experiment. Compare this to the errors in Figure 12 that apply for lossy checkpoint compression.





# 5 Conclusions and Future Work

In the preceding sections, we have shown that using lossy compression, high compression factors can be achieved without significantly impacting the convergence or final solution of the inversion solver. This is a very promising result for the use of lossy compression in FWI. The use of compression in large computations like this is especially important in the exascale era, where the gap between computing and memory speed is increasingly large. Compression can reduce the strain on the memory bandwidth by trading it off for extra computation - this is especially useful since modern CPUs are hard to saturate with low
OI computations.

In future work, we would like to study the interaction between compression errors and the velocity model for which FWI is being solved, as well as the source frequency. We would also like to compare multiple lossy compression algorithms e.g., *SZ*.

*Code availability.* All the code used is available at https://github.com/navjotk/error_propagation . The data used was the Overthrust model from https://wiki.seg.org/wiki/SEG/EAGE_Salt_and_Overthrust_Models

## Appendix A: Additional Results

## A1 Direct compression

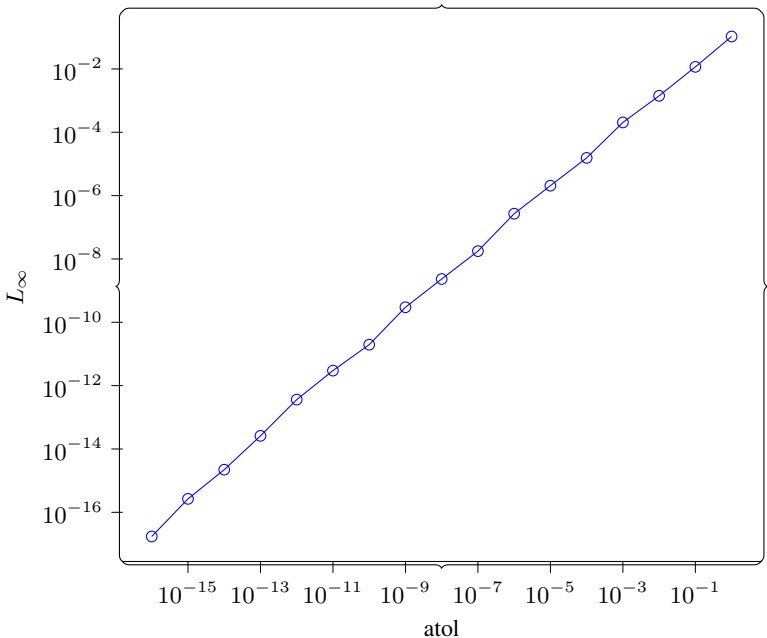

**Figure A1.** Direct compression: $L_\infty$ norm of error versus *atol* . This plot verifies that ZFP respects the tolerance we set.



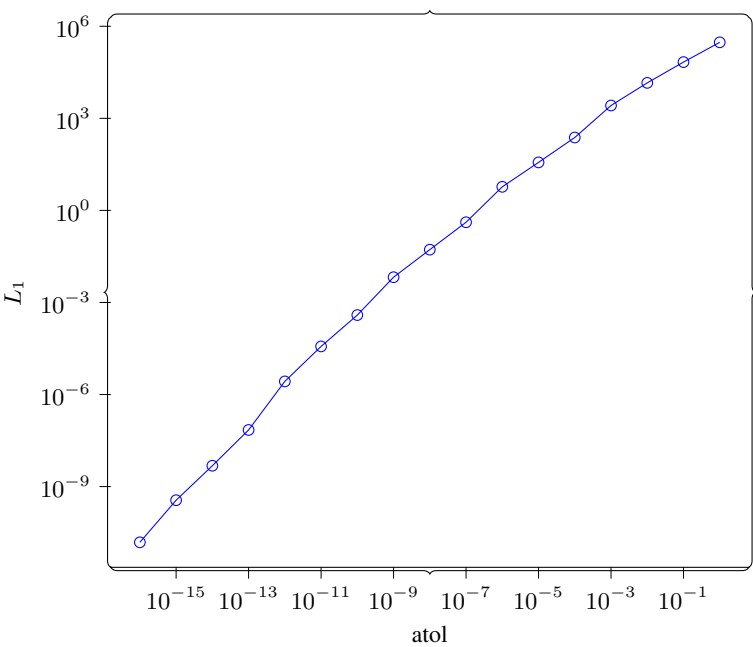

**Figure A2.** Direct compression: $L_1$ norm of error versus *atol* . From the difference in magnitude between the $L_\infty$ plot and this one, we can see how the error is spread across the domain.

## A2 Gradient Computation

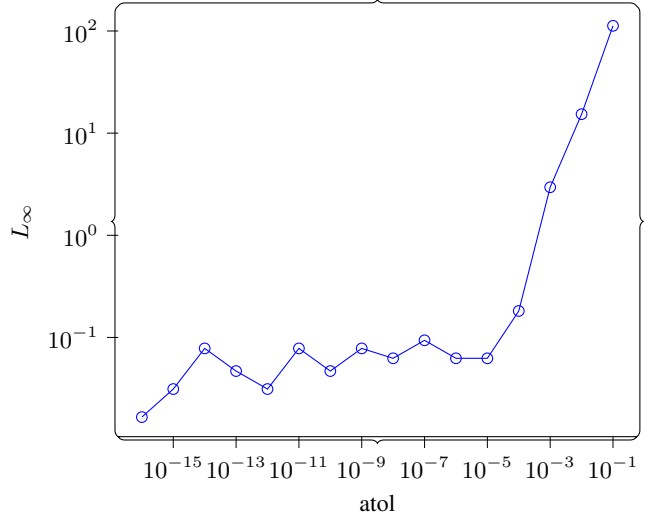

**Figure A3.** Gradient computation: $L_\infty$ norm of gradient error versus *atol* . It can be seen that the error stays almost constant and very low up to a threshold value of $10^{-4}$



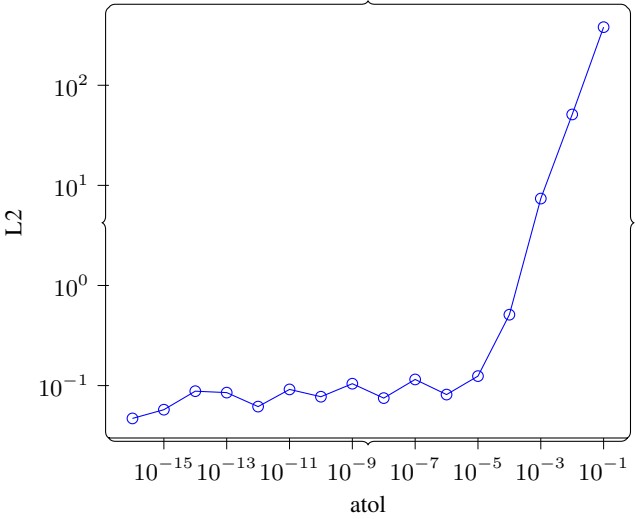

**Figure A4.** Gradient computation: $L_2$ norm of gradient error versus *atol* . It can be seen that the error stays almost constant and very low up to a threshold value of $10^{-4}$

*Author contributions.* Most of the code and experimentation were done by Navjot. The experiments were planned between Jan and Navjot. Jan also contributed in writing. Mathias helped set up meaningful experiments. John contributed in finetuning the experiments and the
presentation of results. Paul and Gerard gave the overall direction of the work. Everybody contributed to the writing.

*Competing interests.* The authors have no competing interests to declare

*Acknowledgements.* This work was funded in part by support from the U.S. Department of Energy, Office of Science, under contract DE-AC02-06CH11357.





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
