# Peer review of "Lossy Checkpoint Compression in Full Waveform Inversion"

_Geoscientific Model Development, 2020_

## Short Comment (SC1) · 14 Nov 2020

Dear authors,

in my role as Executive editor of GMD, I would like to bring to your attention our Editorial version 1.2:

https://www.geosci-model-dev.net/12/2215/2019/

This highlights some requirements of papers published in GMD, which is also available on the GMD website in the 'Manuscript Types' section:

http://www.geoscientific-model-development.net/submission/manuscript_types.html

In particular, please note that for your paper, the following requirements have not been

met in the Discussions paper:

- "The main paper must give the model name and version number (or other unique identifier) in the title."

- "If the model development relates to a single model then the model name and the version number must be included in the title of the paper. If the main intention of an article is to make a general (i.e. model independent) statement about the usefulness of a new development, but the usefulness is shown with the help of one specific model, the model name and version number must be stated in the title. The title could have a form such as, "Title outlining amazing generic advance: a case study with Model XXX (version Y)"."

- "Code must be published on a persistent public archive with a unique identifier for the exact model version described in the paper or uploaded to the supplement, unless this is impossible for reasons beyond the control of authors. All papers must include a section, at the end of the paper, entitled "Code availability". Here, either instructions for obtaining the code, or the reasons why the code is not available should be clearly stated. It is preferred for the code to be uploaded as a supplement or to be made available at a data repository with an associated DOI (digital object identifier) for the exact model version described in the paper. Alternatively, for established models, there may be an existing means of accessing the code through a particular system. In this case, there must exist a means of permanently accessing the precise model version described in the paper. In some cases, authors may prefer to put models on their own website, or to act as a point of contact for obtaining the code. Given the impermanence of websites and email addresses, this is not encouraged, and authors should consider improving the availability with a more permanent arrangement. Making code available through personal websites or via email contact to the authors is not sufficient. After the paper is accepted the model archive should be updated to include a link

to the GMD paper."

In order to simplify the reference to your developments, please add a name (and/or acronym) for your algorithm to the title of your article in your revised submission to GMD.

Additionally, the title should contain a hint which model you used for your publication.

Last but not least, all software used to produce the results shown in the article should be made available in a persistent archive.

Yours,

Astrid Kerkweg

---

## Short Comment (SC2) · 5 Jan 2021

Hello I did not reply to this before as I am still getting used to the submission/review process here and thought I would have the opportunity to respond to everything in one go once all comments have been received. I discovered the "Reply" option today.

Thank you for your feedback. I will make sure to incorporate all your suggestions in the next update of the manuscript.

---

## Referee Comment (RC1) · Anonymous Referee #1 · 17 Jan 2021

Review of manuscript: GMD-2020-325 Title: Lossy Checkpoint Compression in Full Waveform Inversion Authors: Navjot Kukreja, Jan HuÌĹckelheim, Mathias Louboutin, John Washbourne, Paul H.J. Kelly, and Gerard J. Gorman

This paper is a report on using compression for the forward and gradient solutions in full waveform inversion (FWI). They discuss an algorithm that uses compressed storage, as opposed to raw storage, in a checkpoint algorithm in the style of Griewank. A review of some of the existing literature is presented. Then the experimental methodology is discussed. The approach taken is to compare the reference solution (measured in terms of the gradient and forward solution) to the result when the forward solution is compressed. The study is reasonably complete and demonstrates for FWI that compression can be used.

[Figure]

Overall, while the paper has reasonable conclusions I don't think its sufficiently novel to warrant publication. At times it reads like a lab report on using compression technology and I would except any group that would pursue such an approach would have to do a similar exploration. The authors suggest that the idea of using compressed checkpoints is novel, and allows a tradeoff of time and disk space. However, I don't see evidence of exploration of the runtime speeds that could be achieved, and understanding of the accuracy/performance tradeoff beyond the schematic image in Figure 3. Further, the I would expect that this type of approach to be explored elsewhere and be a more minor contribution, particularly since in industry full storage of the forward solution appears to be the norm in FWI. I was hoping that the work would be more generally applicable to PDE-constrained optimization, but due to the use of a linear PDE constraint (which is consistent with FWI) I have my doubts about the extensibility of the results to cases involving turbulent flow, for instance. This is due to the need to linearize the PDE around a state to compute the gradient. Finally, a few oversights for the literature review have been made. The work Weiser and Götschel (See State trajectory compression for optimal control with parabolic PDEs, SISC, 2012) appears to be as complete an empirical study for parabolic systems, with an additional error analysis. I'm not sure how to reconcile this with the statement in the present work that the analysis of compression errors is "beyond known numerical analysis". Overall, I find the present work at best modestly useful, though it reads like a lab report and would be what I would expect every group pursuing such a technology to do. Thus the novelty of the paper, does not warrant publication at this time.

Minor points: 1) In the review, you could cite the growing work in parallel-in-time optimization by for instance S. Gunther or S. Gotschel (going back to the early 2000's Mayday and later Ulbrich and Heinkenschloss). This would likely be in the context of PDE constrained optimization. 2) How is the signal to noise ratio measured? 3) Could there be some more discussion about the wave energy included? It seems that compression would be a dissipative mechanism in general (removing hopefully high frequency modes). How does this impact the objective? Is it systematically less if so

why or why not? 4) The structure of the figures, with one caption per image as opposed to multiple (and some error plots relegated to the appendix) is frustrating to read. This lends itself to the overall feeling that this is a lab report on this material.

—————————————————————

---

## Author Comment (AC1) · 2 Feb 2021

We thank the reviewer for a very considerate review and the insightful comments that have helped us improve the paper. We would be happy to make the following changes in the revised version.

"The authors suggest that the idea of using compressed checkpoints is novel, and allows a tradeoff of time and disk space."

While the reviewer has pointed out relevant prior work in this area, we are not aware of prior work applying lossy compression in combination with checkpointing in seismic imaging. The details of inverse problems vary greatly between different application domains. The tradeoff we discuss is between peak DRAM footprint, total program

runtime, and absolute error tolerance. We do not discuss checkpointing to disk in this paper - the commercial seismic imaging computers are generally designed to have large memory and problems are configured to avoid any checkpointing to disk because the overhead is judged too costly. This is another example of how inverse problems are handled in different sectors.

"However, I don't see evidence of exploration of the runtime speeds that could be achieved, and under- standing of the accuracy/performance tradeoff beyond the schematic image in Figure 3."

We previously studied these aspects in previous work [1], which has been duly cited in the current work. During the previous work, which is generally applicable to adjoint-based optimisation problems, we saw that it is possible to accelerate adjoint computations, depending on the compression factor in the lossy compression. For lossy compression, the compression factor depends on the acceptable error tolerance, and the error tolerance is highly specific to the problem being solved. This current paper is an empirical study of the effect of lossy compression, specific to FWI - i.e. evaluating the impact of lossy compression of the stored wavefield on the accuracy of the computed gradient and overall convergence of the inverse problem.

"Further, I would expect that this type of approach to be explored elsewhere and be a more minor contribution, particularly since in industry full storage of the forward solution appears to be the norm in FWI."

The reviewer is making a reasonable assumption - this may well have been explored in commercial seismic imaging. However, there does not appear to be anything published in the open literature evaluating the impact of lossy compression on seismic inversion. Indeed, a common crude compression technique used in industry and academia is to subsample in time. We have not found any quantitative analysis of this approach in the literature, which is surprising when you see the high level of error incurred using that strategy (see figure 24, section 4.7). The closest work that we are aware of

that performs similar analysis is [2] which performs the cross-correlation (during back-propagation) in Fourier space. That method is implicitly compressive because it only stores a small set of wave numbers. This method has been known about for some time but not frequently discussed quantitatively in literature - this is possibly due to patent CA2760053C on the use of the method in tomographic image acquisition and reconstruction.

"I was hoping that the work would be more generally applicable to PDE-constrained optimization, but due to the use of a linear PDE constraint (which is consistent with FWI) I have my doubts about the extensibility of the results to cases involving turbulent flow, for instance. This is due to the need to linearize the PDE around a state to compute the gradient."

Inverse problems are hard - particularly when dealing with the realities of real data, noise and inexact physics. To optimise the computation of inverse problems (total compute resources and/or time to solution) we have to exploit domain specific knowledge and there is an element of engineering a practical solution. We know that for related problems such as data assimilation in weather/climate science, small perturbations can have a detrimental impact on the solution - in some cases bitwise reproducibility is arguably required. For this reason, we have made it clear in the title and throughout the paper that we are only investigating the use of lossy compression in the domain of seismic imaging which we know to be less sensitive to perturbations compared to chaotic dynamical systems for example.

"The work Weiser and Götschel (See State trajectory compression for optimal control with parabolic PDEs, SISC, 2012) appears to be as complete an empirical study for parabolic systems, with an additional error analysis. "

The paper the reviewer has pointed us at (Weiser and Götschel) is indeed a very valuable addition to the literature review and is certainly relevant. A reference to this paper will be included in the next revision. Weiser and Götschel discusses general parabolic

[Figure]

**[GMDD](https://www.geosci-model-dev-discuss.net/)**

PDEs, while the current work discusses only FWI based on the acoustic wave equation (hyperbolic), going into detail about the propagation of errors through intermediate steps. We also note with interest that Weiser and Götschel devise a new compression algorithm. We focused on use of ZFP as a general-purpose compressor for floating point data as it has the advantage that it's now widely ported. In Section 4.2, Weiser and Götschel talk about compression as an alternative approach to lossy compression. However, by looking at checkpointing and lossy compression in combination, our approach opens up an entire spectrum between these two choices using a tunable cost model.

"I'm not sure how to reconcile this with the statement in the present work that the analysis of compression errors is "beyond known numerical analysis"." We agreed this was phrased poorly and we will correct this in the next version.

"In the review, you could cite the growing work in parallel-in-time optimization by for instance S. Gunther or S. Gotschel (going back to the early 2000's Mayday and later Ulbrich and Heinkenschloss)."

We are aware of the exciting research into exponential time integrators - particularly as applied to weather/climate modelling as a potential way to improve strong scaling while having the additional benefit of necessitating the storage of fewer snapshots during data assimilation. However, to the best of our knowledge, it is still an open research question as to how time varying source (and receiver) terms can be handled in such a formulation when the model time step is near the Nyquist rate required to preserve the signal contained in the (seismic) source terms. If a solution for this mismatch in time scales were resolved for seismic applications then we would see exponential time integration as an additional method to be used in conjunction with these other methods depending on the context. With these points in mind, we currently consider these schemes to be orthogonal to the focus on this paper.

"How is the signal to noise ratio measured?"

We use the following definition for Peak Signal to Noise Ratio (this will be included in the revised version): (Image attached) Where MAX_I is the range of values being compressed (maximum-minimum), and MSE is the mean squared error between the reference solution and the lossy field.

"Could there be some more discussion about the wave energy included? It seems that compression would be a dissipative mechanism in general (removing hopefully high frequency modes). How does this impact the objective? Is it systematically less if so why or why not?"

ZFP's impact on the frequency spectrum of compressed images was previously studied in [3] (see Fig 13, for example). We will add a suitable reference in the paper. It appears from these experiments that compression is actually a benefit. We speculate that lossy compression is acting as a form of regularisation - ie less significant bits are dropped and thereby smoothing effect to the field. As can be seen in [3] the reconstruction errors are normally distributed - so thinking about this as a dissipative mechanism is reasonable. Regularisation is a complicated topic and while we can discuss these observations in the conclusion we are reluctant to make any claims in this regard at this point in our research.

"The structure of the figures, with one caption per image as opposed to multiple (and some error plots relegated to the appendix) is frustrating to read. This lends itself to the overall feeling that this is a lab report on this material."

The figures are in that format as this is what we understand to be required by the GMD submission guidelines for latex - https://www.geoscientific-model-development.net/submission.html#templates "Please provide only one figure file for figures with several panels, and please do not use \subfloat or similar commands."

References [1] Kukreja,N., Hu ÌĹckelheim,J., Louboutin,M., Hovland,P. ,Gorman,G.:Combining checkpointing and data compression to accelerate adjoint-based optimization problems. In: European Conference on Parallel Processing. pp. 87–100.

[Figure]

Springer (2019) [2] Witte, P. A., Louboutin, M., Luporini, F., Gorman, G. J., and Herrmann, F. J.: Compressive least-squares migration with on-the-fly Fourier transforms, Geophysics, 84, 1–76, 2019 [3] Tao, D., Di, S., Guo, H., Chen, Z. and Cappello, F., 2019. Z-checker: A framework for assessing lossy compression of scientific data. The International Journal of High Performance Computing Applications, 33(2), pp.285-303.

―――――――――――――――――

$$PSNR = 10 \cdot \log_{10} \left( \frac{MAX_I^2}{MSE} \right)$$

$$= 20 \cdot \log_{10} \left( \frac{MAX_I}{\sqrt{MSE}} \right)$$

$$= 20 \cdot \log_{10} (MAX_I) - 10 \cdot \log_{10} (MSE)$$

**Fig. 1.**

---

## Referee Comment (RC2) · Anonymous Referee #2 · 25 Mar 2021

The manuscript "Lossy Checkpoint Compression in Full Waveform Inversion" by Kukreja et al. addresses the huge memory requirements of time-domain adjoint simulations that require access to the entire forward wavefield in reverse order. The excellent parallel scalability of time-domain simulations initiated a recent trend to move away from frequency-domain solutions based on the Helmholtz equation, especially on modern hardware using GPU accelerators and for visco-elastic wave propagation. The authors correctly identify the tremendous memory footprint as the major bottleneck in time-domain approaches. Realistic large-scale simulations exceed any reasonable storage capacity by far, making checkpointing and/or wavefield compression approaches inevitable.

I don't know why the industry seems somewhat slow to adopt these techniques, but I assume this is due to using primarily the acoustic wave equation and having a long history in successful inversions using a small set of discrete frequencies, which can be computed in the time-domain with the help of on-the-fly Fourier transforms. However, checkpointing has become state-of-the-art for inverting broadband seismic in continental- to global-scale seismology, and codes like SpecFEM or Salvus provide implementations of it. The authors propose to combine conventional checkpointing with additional lossy compression of the checkpoints to further reduce the memory requirements, which is a neat idea and highly desirable for the above-mentioned reasons.

My major points of criticism are:

- The idea of combining checkpointing and compression itself is not new and has been discussed by the same authors already in a previous publication. Some parts of the manuscript read more like a lab report and some parts are written quite sloppily (see examples in the minor comments below). I am also missing a discussion section that provides a bit of context for the validity of the results for other media. For instance, to what extent does the achievable compression factor depend on the medium, physics, aperture, misfit function, etc.

- I would have assumed that application-tailored compression algorithms would outperform black-box compression tools. Such a comparison would be very interesting and I find it currently missing. Examples of such techniques can be found, for instance, in Boehm et al. (Geophysics, 2016) or Weiser & Gotschel (SISC 2012).

- I am surprised that there is no adaptivity in steering the compression settings throughout the inversion. From an optimization perspective with inexact derivatives, it is well-known that more accurate gradients would be required when approaching a stationary point. I can't find this mentioned or analyzed in the manuscript.

- The manuscript would really benefit from a 3D visco-elastic example as this is where the memory bottleneck becomes a lot more prominent. Of course, I am not asking

for a 3D FWI example, but I would strongly encourage the authors to provide an error analysis for a single visco-elastic gradient computation in 3D.

A few more minor comments:

- You reference eq. (1) at least twice already in the introduction, way before the equation is actually introduced. Similar premature references exist for eq. (4) and symbols like Phi(m).

- I also don't think this should be focusing on the non-dissipative acoustic wave equation. The compression approach applies to all time-dependent wave equations and it is a lot more relevant for visco-elastic media. You already hint that eq. (1) is not really the equation of interest on p.3, line 48.

- I think the statements on p.3 lines 50-56 are misleading and / or incorrect. The communication overhead of MPI-parallelized simulations can well be hidden behind computations by computing the halo first and performing asynchronous communication. But even on distributed compute architectures reducing the memory footprint is highly desired. Furthermore, many frequency-domain methods indeed have a HUGE memory footprint when factorizing the Helmholtz operator. At least, you would need to be more specific what you mean by frequency-domain here.

- p.5, line 106. Typo: There is an additional ")" in (2016)).

- Section 2. The notation mixes bold and italic symbols, for instance, for the model m and d_obs / d_sim, phi... The operator A is not introduced properly. It is the discretization of the PDE operator, and not of the equation.

- I would consider merging sections 3 and 4, because section 3 merely contains extended headings for the subsections of part 4.

- Many figures contain similar quantities and could be merged. For instance, Fig 7/8, Fig 12/13, Fig 14/15, Fig 20/21 could all be merged into a single row each.

- The maximum buffer size used in Fig 7 and 8 is fairly small and I would consider extending the line to at least twice the number of time steps.

---

## Author Comment (AC2) · 14 Apr 2021

We thank the reviewer for the insightful comments. Every one of these points is helping make this paper much stronger.

Major points:

- *The idea of combining checkpointing and compression itself is not new and has been discussed by the same authors already in a previous publication. Some parts of the manuscript read more like a lab report and some parts are written quite sloppily (see examples in the minor comments below). I am also missing a discussion section that provides a bit of context for the validity of the results*

*for other media. For instance, to what extent does the achievable compression factor depend on the medium, physics, aperture, misfit function, etc.* We can rephrase the contributions section to not claim that the idea itself is new. This is an empirical study and we can only carry out a limited number of experiments. While we would have liked to have studied multiple media, physics, aperture and misfit functions, we have left this as future work and mentioned it as such. Happy to add a few lines to make these limitations of our study clearer.

- *I would have assumed that application-tailored compression algorithms would outperform black-box compression tools. Such a comparison would be very interesting and I find it currently missing. Examples of such techniques can be found, for instance, in Boehm et al. (Geophysics, 2016) or Weiser Gotschel (SISC 2012).* The two studies the reviewer mentions are both very relevant. Boehm et al is already cited as such, and Weiser and Gotschel is on the list of planned improvements for the next revision after comments from Reviewer 1. We would be happy to add a discussion about how different compression algorithms might affect the trade-off between accuracy, runtime and peak memory.

- *I am surprised that there is no adaptivity in steering the compression settings throughout the inversion. From an optimization perspective with inexact derivatives, it is well-known that more accurate gradients would be required when approaching a stationary point. I can't find this mentioned or analyzed in the manuscript.* Thanks for pointing this out. Our method takes atol as an input and not mentioning where that atol comes from is a clear oversight. We will add a couple of lines about this kind of adaptivity that we are enabling with our method. We would also like to point out that there is a level of adaptivity already present that we have left out from the paper for simplicity. The whole inversion problem presented in our paper is one instance of a multigrid method that starts off by inverting on the coarsest possible grid with the lower frequencies, and then slowly adding higher frequency components and interpolating onto finer grids over multiple inversions. We will also discuss this level of adaptivity in the next revision of the paper.

- *The manuscript would really benefit from a 3D visco-elastic example as this is where the memory bottleneck becomes a lot more prominent. Of course, I am not asking for a 3D FWI example, but I would strongly encourage the authors to provide an error analysis for a single visco-elastic gradient computation in 3D.* The memory bottleneck becomes more prominent as we move to more complex physics requiring multiple fields. There is also more computation to do, of course. This can be seen as another data point on the "arithmetic intensity" axis. We would be happy to add an example of how the tradeoff changes with different physics - potentially a plot along this arithmetic intensity axis. However, as we show in Table 1, the memory bottleneck is already significant for the isotropic acoustic equation. Also, as far as we are aware, isotropic-acoustic FWI is more common in practice than visco-elastic FWI as of today. Hence we don't think that isotropic-acoustic is irrelevant or too small an example to be interesting.

Minor comments:

- *You reference eq. (1) at least twice already in the introduction, way before the equation is actually introduced. Similar premature references exist for eq. (4) and symbols like Phi(m).* We would be happy to rewrite these sections to fix this in the next revision.

- *I also don't think this should be focusing on the non-dissipative acoustic wave equation. The compression approach applies to all time-dependent wave equations and it is a lot more relevant for visco-elastic media. You already hint that eq. (1) is not really the equation of interest on p.3, line 48.* As discussed above, we chose to focus on the isotropic acoustic case because that is the more commonly used case in practice. We agree that any memory-saving technique becomes

more relevant when the equation of interest has more fields. The change previously discussed should increase the applicability of this work beyond isotropic acoustic.

- *I think the statements on p.3 lines 50-56 are misleading and / or incorrect. The communication overhead of MPI-parallelized simulations can well be hidden behind computations by computing the halo first and performing asynchronous communication. But even on distributed compute architectures reducing the memory footprint is highly desired. Furthermore, many frequency-domain methods indeed have a HUGE memory footprint when factorizing the Helmholtz operator. At least, you would need to be more specific what you mean by frequency-domain here.* About the frequency-domain methods, we are happy to rephrase this to be clearer. About MPI, we are already accounting for computation-communication overlap in what we say. The point we are trying to make is that there is a lower limit to the amount of computation required to hide the communication behind. The communication time depends on the interconnect - which is not always very fast on cloud platforms, so the smallest per-rank subdomain that would be able to hide the communication behind its in-domain computation might be too big on a system without a high-speed interconnect. Happy to expand this discussion in the next revision of the paper.

- *p.5, line 106. Typo: There is an additional ")" in (2016)).* Will be fixed in the next revision.

- *Section 2. The notation mixes bold and italic symbols, for instance, for the model m and d_obs / d_sim, phi. The operator A is not introduced properly. It is the discretization of the PDE operator, and not of the equation.* Will be fixed in the next revision.

- *I would consider merging sections 3 and 4, because section 3 merely contains extended headings for the subsections of part 4.* We think Section 3 provides

value as a reference to understand what each experiment does, in a quick read that typically goes back and forth through the paper. Happy to remove if the reviewers think this is creating confusion.

- *Many figures contain similar quantities and could be merged. For instance, Fig 7/8, Fig 12/13, Fig 14/15, Fig 20/21 could all be merged into a single row each.* Happy to fix

- *The maximum buffer size used in Fig 7 and 8 is fairly small and I would consider extending the line to at least twice the number of time steps.* We understand the reviewer to mean that we should run this experiment for more timesteps than 300. Happy to update this plot to twice as many timesteps in the next revision.

---

## Author Response (AR1)

We thank the reviewers for very thorough review and the insightful comments that have helped us improve the paper. We have previously written point-by-point responses to all reviews received in the interactive discussion. We have made the following changes in this revision:

1. Changed the title to be more specific
2. Removed most forward references in the text – there is one remaining, which we believe improves readability.
3. Expanded the description of why domain-decomposition does not solve all peak memory problems.
4. Reference added to Weiser and Götschel (State trajectory compression for optimal control with parabolic PDEs, SISC, 2012).
5. "I'm not sure how to reconcile this with the statement in the present work that the analysis of compression errors is "beyond known numerical analysis". Rephrased.
6. Some rephrasing to make it clear that the key idea of Lossy compression of checkpoints was previously presented by us in a more general context of adjoint computations. This is a followup work that discusses application-specific error tolerances.
7. Fixed notation for consistency.
8. Added a discussion of the adaptivity already present in the method.
9. Added section on Error metrics where PSNR and other metrics are defined.
10. Increased the number of timesteps in the forward propagation experiment.
11. Combined some figures into multipane figures.
12. Added a discussion section that discusses the validity of these results beyond what was studied.
13. Code and Data references are all DOIs now.

---

## Author Response (AR2)

We thank the reviewer for the in-depth feedback that has helped improve this paper. We are happy to make changes in response.

*Several figures (8, 10, 11, 12, 14, 16) are not explained or even referenced in the text.*
Text has been added to reference and explain all these figures.

*How do you manage parallel file formats for distributed simulations with compressed chunks of data, where the size of each chunk might vary and unknown prior to the simulation?*
The compression/checkpointing discussed in the paper is all in memory, and not the filesystem. After initial application startup, the only filesystem access is in reading shot data, and that is not a bottleneck in the setup we studied. We added text to the paper making this more explicit, along with justification. Due to this being in-memory, the distributed nature of the computation does not affect the discussion in the paper since each rank is only concerned with its own part of the domain.

*Did you compare different compression algorithms, and can you comment on the computational overhead for (de)compressing the wavefield?*
These questions were addressed in our previous publication on this topic [1]. We added text to the paper referring the interested reader to this.

*Which fields do you compress (pressure, velocity, pressure gradient, …)? Do you apply different tolerances or compression strategies for different fields?*
Since we are solving the acoustic isotropic equation, there is only one field present - the pressure field, and we compress it directly i.e. not the gradient. We have added text making this clearer.

*Is it possible to a-priori ensure an absolute / relative tolerance after decompression?*
This is the definition of atol (absolute tolerance), for ZFP, our chosen compressor for this work. We have added text making this clearer.

*Can you elaborate more on the absolute tolerance atol used in the numerical examples? I think it would be better to somehow relate the tolerance to the maximum amplitude of the wavefield / resp. source. The absolute number is rather meaningless.*
Figure 7 (PSNR vs atol) is intended to relate a relative error metric (PSNR) to the chosen absolute error tolerance.

*When looking at Fig. 7, I am wondering if the frequency content of the (de)compressed snapshot is altered, which is why the error is growing in the first couple of time steps?*
We have not studied the frequency content of the decompressed snapshot in this work. Partially because this was previously studied in [2]. This is a work we cite in our paper. Also, after changing the plot to PSNR (to address the reviewer's next point), we believe that the initial increase in error might not be significant in the bigger picture.

*Furthermore, is the decreasing trend a simple result from the decaying amplitudes in the wavefield or is this normalized in some way? It would help to also show relative errors per time step.*
We have changed the forward error plot to show PSNR to address this.

*+++ Minor comments*

*page 2, line 22:*
*Referencing an equation long before it appears in the manuscript is bad style.*
Reference removed

*Furthermore, the equation is not called TTI. TTI refers to the medium / model parameterization or stress-strain relation, respectively, and should not be used as an acronym without introduction.*
We now refer to it as "the acoustic anisotropic equation in a Tilted-Transverse Isotropic medium"

*page 2, Table 1:*
*"Forward propagation" is misleading and should be "time steps" instead.*
Text changed

*Calling it "peak memory" is very misleading for gradient computations because no reasonable implementation would do that.*
We understand that no reasonable implementation would do this because this is an unreasonable amount of memory, and practical implementations would employ compromises on runtime and accuracy to make it more reasonable. This is also the point we are trying to drive with that table - that we are proposing one new way of avoiding that exorbitant memory consumption.

*page 3, line 48/49:*
*Either remove the reference to eq. (1) or state the equation here.*
Reference removed.

*page 4, line 81:*
*Why is there an asterisk after Louboutin?*
Removed

*page 5, Figure 2:*
*Add labels and annotation to make it easier to read. The horizontal axis could count multiples of single simulations.*
Figure updated

*page 9, line 191:*
*What density model are you using? I would still consider it an inverse crime if it were a scaled version of the velocity model or even just a homogeneous model.*
We have used the Gardner relationship to derive the density model.

*section 3:*
*This section seems a bit disconnected from the rest. I would recommend merging it with section 4. In particular, I don't see a reason to introduce the subsections of section 4 already here with a single paragraph.*
The part of section 3 in question has been merged into section 4.

*page 11, Figures 4 and 5:*

*How does the absolute tolerance of 1e-4 relate to the pressure amplitude? Which other fields do you compress is this really an absolute tolerance and not a relative one?*
We have added a note to the plot of the wavefield to highlight that the scale shown is of thresholded values, to aid visualisation. The low values in that plot might be contributing to the reason this question came up.
We have also added a histogram of pressure values to answer questions about the pressure amplitude.
The number we call absolute tolerance is the number we provide as input to ZFP in its field called "absolute tolerance" - which represents the absolute value of the error tolerance we are willing to incur as part of the lossy compression. We specify this at the point of compression. The actual errors incurred might, of course, be smaller. Figure A1 helps to compare actual errors incurred vs provided absolute tolerance values.

*page 12, caption Figure 6:*
*There is a reference missing: "See figures A1 and ??"*
Fixed

*page 12, Figure 6:*
*How do you define the signal to noise ratio in this case?*
This is defined in Section 3.2

*page 15, Figure 10:*
*The same results are shown again in Fig. 19, so I would either show the entire x-axis here or remove the figure.*
Figure updated.

*page 15, Figure 11:*
*What error is shown on the y-axis? And why is it huge?*
This figure shows the Taylor-series linearization error in the FWI objective function when using the calculated gradient for linearization. Since the absolute value of FWI objective function in the setup used is so high, these errors, which are actually miniscule on the relative scale, appear much larger. The focus of the figure was not supposed to be on the magnitude (absolute or relative) of these errors - but on the fact that the lossy gradient shows the same behaviour as the reference gradient (there are two curves in that plot - perfectly coinciding). Upon reflection, we feel that this figure is not serving its purpose and only creating confusion since the error shown here is very different in nature from all the other error plots, and is to be interpreted differently. Since this is such a tangent to the flow of the paper, we feel it's best to remove this figure.

*page 17, line 269:*
*Please put atol in context to the maximum amplitude of the wavefields that are compressed.*
We have added a histogram of pressure values to make this easier.

*page 17, Figure 14:*
*Should this be "true model" instead of "true solution"? The figure is not referenced in the text.*
Text updated to call it true model as well as reference the figure.

*page 17, Figure 15:*

*Some of the recovered structure looks to be significantly smaller than a wavelength for the given frequency content. Could you comment on inverse crime? Are you inverting for density as well (see question on the density model above)?*
The resolution of FWI is about 1/10th of the peak wavelength so the result is within the expected resolution

*page 20, line 293:*
*What does atol > 4 mean?*
Text updated to atol > $10^{-4}$

[1] Kukreja, N., Hückelheim, J., Louboutin, M., Hovland, P. and Gorman, G., 2019, August. Combining checkpointing and data compression to accelerate adjoint-based optimization problems. In *European Conference on Parallel Processing*(pp. 87-100). Springer, Cham.

[2] Di, S., Tao, D., Liang, X., Li, S., Chen, Z. and Cappello, F., Z-checker (0.1. 3) Compression Assessment Report for EXAFEL.

---

## Author Response (AR3)

The following changes were made to the manuscript since the last revision:

1. Added City and Country names to all author affiliations, as asked.
2. Added line in Acknowledgements thanking the reviewers.
3. Fixed some minor grammatical errors that were found in a final proofread.